# Electroless Plating of High-Performance Composite Pd Membranes with EDTA-Free Bath

**DOI:** 10.3390/ma14174894

**Published:** 2021-08-27

**Authors:** Jun-Yi Wang, Yen-Hsun Chi, Jin-Hua Huang

**Affiliations:** 1Department of Materials Science and Engineering, National Tsing Hua University, Hsinchu 30013, Taiwan; jeff_ohno@hotmail.com; 2Green Energy and Environment Research Laboratories, Industrial Technology Research Institute, Hsinchu 31040, Taiwan; ansonchi@itri.org.tw

**Keywords:** palladium, electroless plating, hydrogen separation, membrane modification

## Abstract

High-performance composite Pd membranes were successfully fabricated using electroless plating with an EDTA-free bath. The plating started with employing the one-time addition of hydrazine. In the experiment, the hydrazine concentrations and plating bath volumes were systematically varied to optimize the plating. The optimum composite Pd membrane tube showed high H_2_ permeance of 4.4 × 10^−3^ mol/m^2^ s Pa^0.5^ and high selectivity of 1.6 × 10^4^, but poor cycling stability. Then, a method of sequential addition of the hydrazine from the high to low concentrations was employed. The resultant membrane, about 6 μm thick, still exhibited a high selectivity of 6.8 × 10^4^ as well as a much-improved plating yield and cycling stability level; this membrane outperformed the membrane made using the unmodified plating technique with the EDTA-contained bath. This result indicates the EDTA-free bath combined with the sequential addition of hydrazine is a simple, low-cost, yet effective method for preparing thin, dense composite Pd membranes featuring high hydrogen permeation flux and high thermal durability.

## 1. Introduction

With the diminishing stock of fossil fuels and the growing concerns regarding environmental pollution problems, scientists have been making a great effort to search for renewable and clean energy resources. Because hydrogen fuel cells generate no pollutants and generally exhibit better efficiency than conventional internal combustion engines [1], hydrogen has been considered as one of the solutions for energy and environmental issues. Hydrogen can be obtained by cooperating water electrolysis with renewable power generation systems, such as solar and wind power [2,3]. It is attractive because no pollutants would be generated during hydrogen production, but the higher cost of this technique compared to hydrogen production from fossil fuels and biomass still limits its usage [4,5]. At present, the fossil fuel reforming method still comprises the majority of hydrogen production [6]. Because the cost of hydrogen production via fossil fuels heavily relies on the fuel price, areas with low fuel costs, such as the Middle East and North America, have much lower hydrogen production costs. Extracting hydrogen from fossil fuels and biomass will inevitably generate by-product gas, such as carbon monoxide, carbon dioxide, and methane. Notably, carbon monoxide would poison fuel cells and degrade cell performance; thus, further purification of as-produced hydrogen is needed. The development of effective methods of hydrogen separation, therefore, has raised considerable interests within academia and industry [7].

Among the various types of hydrogen separating inorganic membranes, metal membranes made of palladium (Pd), and its alloys are the most widely studied due to their unique permselectivity to hydrogen [8,9]. Theoretically, ultra-pure hydrogen can be obtained from defect-free Pd. However, in order to lower the cost of Pd and shorten the diffusion length (thus resulting in higher H_2_ permeance), many researchers have focused on the fabrication of composite Pd-based membranes, which generally consist of a Pd-based thin film with a thickness of several micrometers on porous substrates, such as porous stainless steel (PSS) and microporous glass tubes [10,11]. Several methods have been developed to fabricate the composite Pd-based membranes, including sputtering, chemical vapor deposition, electroplating, and electroless plating [12,13,14,15,16,17]. Among these methods, electroless plating is attractive because it uses only simple equipment that is easy to scale up.

Electroless plating is a well-developed technique that is widely used to prepare dense Pd membranes on porous substrates. Procedures of electroless plating in recent research are normally similar. Indeed, recent research on Pd membrane preparation mostly focuses on other aspects, especially the intermediate layer between the Pd and the substrate [18,19,20]. Few researchers put effort into plating bath modification because it is a fairly mature technique. Conventional plating baths used in electroless plating contain a chelating agent, EDTA (ethylenediaminetetraacetic acid). to control the plating rate and prevent the whole plating stage from bulk decomposition. However, a trace of the EDTA tends to remain in the as-fabricated membranes. Volpe et al. [21], for example, detected an EDTA signal in an EDTA-contained plating, bath-derived Pd membrane by using Fourier transform infrared spectroscopy. Roa et al. [22] also found EDTA residue through thermogravimetric analysis. EDS analysis can also point out carbon impurities inside the Pd membrane derived from EDTA-contained bath [23].

Unfortunately, incorporated EDTA in membranes not only could serve as stress that produces defects, but it could also evolve into carbon monoxide at high temperatures and cause degradation of the membrane performance thereafter. In this regard, several researchers have successfully fabricated palladium membranes with an EDTA-free bath. Gade et al. [24], for example, fabricated unsupported Pd membranes from an EDTA-free bath at 50 °C. However, the relatively high plating temperature of 50 °C made the plating unstable so that the plating stage could last only 20 min. As a result, eight or more plating stages were needed to achieve a visibly pinhole-free membrane. Moreover, the repeated short plating cycles resulted in low-yield Pd membranes and made the fabrication costly. Ryi et al. [25] fabricated Pd membranes on porous Hastelloy discs with an EDTA-free bath at a lower working temperature of 25 °C. Nevertheless, the relatively small surface area of Hastelloy discs would limit its gas flow in practical use. Thoen et al. [26] fabricated Pd-Cu membranes onto alumina tubes with an EDTA-free bath, and the results showed that the nitrogen flux drastically increased after six days of permeation testing. Although carbon contamination was absent, the use of EDTA-free baths in electroless plating still needs further investigation for practical use. Based on the aforementioned references, plating without EDTA would be relatively tough to execute. Parameters, such as plating time or plating temperature, should be tuned and monitored to achieve a high-quality Pd membrane.

In this work, thin Pd membranes were fabricated on the PSS tubes using electroless plating with an EDTA-free bath, as well. The PSS tubes were chosen to be the substrate materials because of the concern about practicality and cost reduction. Initially, hydrazine concentration and plating bath volume were varied. For each plating condition (a set of hydrazine concentration and plating bath volume), the tube would undergo two identical plating stages. Performances of different tubes were then discussed in detail to find the optimized plating parameters. However, all as-fabricated membranes showed poor thermal stability. To solve for this, the sequential addition of hydrazine with concentrations from high to low was introduced. The resultant membrane, prepared using the EDTA-free bath and sequential addition of hydrazine. showed a competitive permselective performance, high plating yield, and much-improved reliability.

## 2. Materials and Methods

### 2.1. Membrane Preparation

Hollow PSS tubes, as supporting substrates for loading Pd membranes, were purchased from Pall Corporation (New York, NY, USA). The PSS tubes had an outer diameter of 11.8 mm, a wall thickness of 0.89 mm, and a length of 150 mm. To fit the gas flux measuring module, a nonporous stainless-steel tube was welded to one end of each PSS tube, and a nonporous stainless steel cap was welded to the opposite end. Then, the welded PSS tubes were ultrasonicated sequentially in alkaline, isopropyl alcohol, and deionized water to remove surface contaminants, such as oil, dirt, and grease. Finally, the PSS tubes were dipped in an isopropanol bath and then dried at 393 K overnight.

The commercial PSS tubes usually display high surface roughness with large and on-uniform pores, which could lead to cracks, pinholes, and defects over the surface of as-fabricated membranes. Therefore, a two-step surface modification was undertaken to reduce the pore size of the PSS surface. Details of the surface modification procedures have been described in our previous studies [16,17]. In brief, 10 μm, 99.5% alumina (Al_2_O_3_) particles purchased from Merck (Burlington, MA, USA) were first used to fill the large pores of the PSS substrate surface. Then, the Al_2_O_3_/PSS tubes were immersed in solutions with a lithium–aluminide ingot (80/20 wt%) intermetallic compound (Li-Al IMC) (99.9%, UMAT, Hsinchu, Taiwan) to ensure the growth of the Li-Al layered double hydroxide (LDH) layer and further reduce the pore size. Both the Al_2_O_3_ particles and the LDH layer not only smoothed the surfaces of the PSS tubes, but also served together as the diffusion barrier between the PSS tube and Pd membrane.

After surface modification with Al_2_O_3_ and LDH, a surface activation process was carried out to ensure better adhesion of the Pd during the plating stage. Each LDH-Al_2_O_3_-modified PSS tube was subjected to a surface activation process, which was carried out by alternately immersing the PSS tubes into the SnCl_2_ (99.7%, Macron, Radnor, PA, USA) and PdCl_2_ (99.99%, UR, New Taipei, Taiwan) solutions several times. After activation, the PSS tube surface was expected to be fully covered with a Pd nuclei layer.

Pd membranes were deposited onto the PSS tubes at 30 °C with an EDTA-free bath. The EDTA-free bath was composed of PdCl_2_, HCl, and NH_4_OH, and was stirred well to form a fine mixture before electroless plating. The detailed composition of the EDTA-free bath is described in Table 1. During plating, the PSS tubes were immersed into glass tubes with an EDTA-free bath. Plating temperature was controlled by a larger water tank. Throughout each of the plating stages, tubes would be rotated at a fixed speed to achieve a smoother membrane. After that, the plating with an EDTA-free bath was first conducted by undertaking two identical plating stages; each stage lasted 90 min and included the one-time addition of hydrazine (N_2_H_4_), as depicted in Figure 1a. Five concentrations of hydrazine (4.5, 6.75, 9, 13.5, and 18 mM) and three volumes of plating bath (120, 180, and 240 mL) were systematically varied to optimize the plating. Then, the approach of using the one-time addition of the hydrazine was changed to sequential addition, wherein hydrazine was sequentially added—in concentrations from the high to low—to the same plating bath, as demonstrated in Figure 1b. The membranes fabricated from the EDTA-free bath that utilized the sequential addition of hydrazine have shown excellent H_2_ permselectivity and thermal durability.

### 2.2. Characterization

The hydrogen permeance and nitrogen leakage rate were measured with a homemade gas flux measuring module, as schematically shown in Figure 2. The temperature was initially ramped to 400 °C at a rate of 1 °C/min with nitrogen flowing, and then it was kept at 400 °C for 5 h with flowing hydrogen to undergo surface reduction. After that, both hydrogen and nitrogen fluxes were measured at 1–4 bar pressure differences. Cycling stability tests, including temperature rising and descending on a daily basis, were carried out afterward to verify the thermal durability of the membranes. Permeation measurements were undertaken daily to realize the rate of performance degradation. Both gases used in the permeation test had a purity level of 99.99% or higher. Moreover, all permeation tests were conducted in a single-gas situation; the results would not be typical of a real case of mix-gas. The morphology of as-prepared Pd membranes was obtained through scanning electron microscopy (SEM, JSM-6500F, JEOL, Tokyo, Japan). The tubes for SEM analyses were cut into small pieces and embedded in epoxy. The embedded tubes were then ground and polished to obtain clear SEM images. The membrane thickness measurements were estimated based on the cross-sectional SEM micrographs and a weight-gain method, which showed consistent results.

## 3. Results and Discussion

In this work, the palladium film during electroless plating with an EDTA-free bath glowed according to the following reaction:2 Pd^2+^ + N_2_H_4_ + 4 OH^−^ → 2 Pd + N_2_ + 4 H_2_O(1)

As Equation (1) shows, the N_2_H_4_ concentration influences the reaction rates of the as-fabricated Pd films. Likewise, the volume of the plating bath determines the Pd content and film thickness, thereby strongly influencing the morphology of Pd membranes. Initially, N_2_H_4_ was added in the bath using the one-time addition approach, as depicted in Figure 1a. In the experiment, the two parameters, namely the volume of the plating bath and the concentration of N_2_H_4_, were systematically varied to optimize the EDTA-free bath. The surface area of the PSS supports in this study was about 61 cm^2^, and the volumes of plating bath were chosen to be 120, 180, and 240 mL, thus yielding the corresponding volume–to–surface area ratios of approximately 2, 3, and 4, respectively. Table 2 summarizes the properties and performances of as-fabricated Pd membranes. Be aware that the H_2_ permeance in Table 2 was obtained by plotting the H_2_ flux against the difference of the square root of pressure. The coefficient of determination, or R^2^, is greater than 0.999 for all membranes listed in the table, which means the H_2_ permeation in this study is basically under the diffusion-controlled mechanism.

It is obvious that for the three concentrations of N_2_H_4_ investigated, the membrane thickness increased as the bath volume increased. This result is reasonable because a higher volume of bath, which corresponds to a higher content of palladium ions, could sustain a relatively stable reaction rate for a longer plating time. However, the hydrogen permeance decreased as the bath volume increased, which could be attributed to the increasing thickness of the membrane, as H_2_ permeation has been previously proved to be a diffusion-controlled process. Figure 3 shows the surface morphologies of the Pd membranes fabricated using the same 6.75 mM of the N_2_H_4_ addition but different bath volumes of 120 and 240 mL. For the membrane fabricated from the 120 mL bath (shown in Figure 3a), a relatively rough surface was observed. However, the vertical, sheet-like structure of the LDH morphology could still be distinguished, revealing that the Pd membrane in Figure 3a is relatively thin and thorough coverage of the Pd is not yet formed. The high N_2_ flux and low H_2_/N_2_ selectivity measured from this sample confirm the observation. On the contrary, the membrane obtained from the 240 mL plating bath (shown in Figure 3b), shows a much smoother surface without evident pinholes. This morphology difference could explain why the latter one has a much lower N_2_ leaking flux.

Mardilovich et al. have shown that there is a minimum Pd film thickness required to form a dense membrane on a porous substrate using electroless plating, which is about three times the size of the largest pore of the porous substrate [27]. In this study, the threshold thickness was found around approximately 3.5–4 μm. It is relatively tough to form a dense membrane via the 120 mL bath because of a lower content of Pd^2+^ in the plating bath.

As Table 2 states, both the bath volume and the N_2_H_4_ concentration can influence the resultant thickness of the Pd. Unlike bath volume, an increased concentration of N_2_H_4_ can raise the reaction rate and hence increase the resultant thickness. The calculated plating yield in Table 2 also confirms this. According to Yeung et al., the plating rate of the N_2_H_4_-based baths should be correlated with the concentration of N_2_H_4_ [28]. Furthermore, Yeung et al. also pointed out the larger grains of palladium could be obtained with higher concentrations of the N_2_H_4_ addition.

In this study, five different concentrations of the N_2_H_4_ addition, ranging from 4.5 to 18 mM, were applied. For the lowest concentration (i.e., 4.5 mM) of the N_2_H_4_ addition, the nitrogen leakage rates of the as-fabricated membranes are, generally, quite high. This can be ascribed to the relatively thin, porous membranes obtained. Similarly, the membranes fabricated from the highest concentration (i.e., 18 mM) of the N_2_H_4_ addition also show relatively higher nitrogen leakage rates, although the thickness levels are far beyond 5 μm, which implies these thick membranes are not as dense as expected. The plating baths of these membranes were found to turn murky during plating. Moreover, several large particles, in the range of several millimeters, could be visually observed on the surface of the membranes after plating. Both phenomena indicate a bulk decomposition of Pd during plating for the case of the 18 mM N_2_H_4_ addition. Consequently, the concentration of an N_2_H_4_ addition should be within the range of 6.75 to 13.5 mM to achieve a palladium membrane with better performance.

To examine whether the as-made membranes are practical in use, the membranes made from the 6.75 and 9 mM N_2_H_4_ additions (with the same bath volume of 180 mL) were selectively chosen to undergo the cycling stability test. These results are shown in Figure 4. Although the two membranes exhibited different selectivity values (1.6 × 10^4^ and 5.0 × 10^3^, respectively) in the beginning, they deteriorated afterward, with selectivity steadily dropping to about 2000. Since the N_2_ flux of both membranes increased steadily throughout the test, the membranes are expected to fail during longer operation times. Moreover, the plating yield of qualified membranes was relatively low. To solve these two problems, we induced a multi-step method of plating, which consisted of stages with plating rates from high to low, as demonstrated in Figure 1b. The arrangement of the plating rate was achieved by controlling the concentrations of hydrazine. Based on the research of Felix’s group, pore-clogging is more easily achieved through a faster plating rate, which can be schematically illustrated in Figure 5 [29]. In this experiment, a high concentration of the hydrazine addition in the first plating stage was used to ensure that pore-clogging took place in a relatively short amount of time. This could ensure most of the Pd deposits were on the surface instead of being in the pore of the PSS tube. After the fast-plating stage, three slow plating stages, with lower concentrations of the hydrazine addition, were employed to fabricate a dense membrane with fewer defects and a greater plating yield.

Table 3 compares the properties and performances of the Pd membranes that were prepared using the two different approaches of N_2_H_4_ addition with EDTA-free baths. As mentioned previously, a threshold thickness of the Pd film is required to obtain a dense, defect-free membrane. Therefore, changing the plating bath was inevitable, and the plating yield was reduced accordingly for the one-time addition approach. On the contrary, by using the new approach of sequential addition of the N_2_H_4_, changing the bath was no longer needed and the plating yield could be greatly increased. In Table 3, the membrane prepared through sequential addition exhibited a high H_2_/N_2_ selectivity and a drastically enhanced plating yield, though with slightly reduced hydrogen permeance. The cycling stability test of the membrane, prepared using sequential addition, was also undertaken. As can be seen in Figure 6, the H_2_ and N_2_ fluxes varied similarly with the daily test. The two fluxes first increased soundly and then increased slowly as time passed, indicating an almost stable selectivity of the membrane. The result indicates that the as-fabricated membrane can work for a longer time, which means the new approach for the hydrazine addition is more practical in use.

## 4. Conclusions

In conclusion, we have presented a systematic study on the electroless plating of Pd membranes on porous stainless-steel tubes using an EDTA-free bath. Initially, the plating was conducted by varying the bath volume and N_2_H_4_ concentration using the one-time addition of N_2_H_4_. We found that a bath volume of 180 to 240 mL and an N_2_H_4_ concentration of 6.75 to 13.5 mM were appropriate to support the formation of Pd membranes with better performances. With a small volume of bath, the Pd film thickness would be too small to form a dense membrane; however, with a large volume of bath, the hydrogen permeance of the membrane would be greatly decreased because of the increasing thickness. On the other hand, a low concentration of the N_2_H_4_ addition would result in a membrane with large nitrogen flux owing to the small thickness; conversely, a high concentration of the hydrazine addition would cause bath decomposition, leading to a membrane with large nitrogen flux, too. The optimum composite membrane tube was obtained from the 180 mL bath and 6.75 mM hydrazine addition, which exhibited H_2_ permeance of 4.4 × 10^−3^ mol/m^2^ s Pa^0.5^ and selectivity of 1.6 × 10^4^. However, all as-fabricated membrane tubes degraded quickly upon the temperature cycling test, which limited their practical use. An arranged concentration sequence of N_2_H_4_ additions from high to low was then introduced. The fast-plating rate in the first stage ensured pore-clogging, while the slow plating rate afterward helped reduce the defects to the membrane. The resultant membrane thus showed much better stability and higher plating yield. The deposition method, the EDTA-free bath together with the sequential addition of N_2_H_4_, is simple and low-cost, yet effective, which made it a competitive technique in terms of practical use.

## Figures and Tables

**Figure 1 materials-14-04894-f001:**
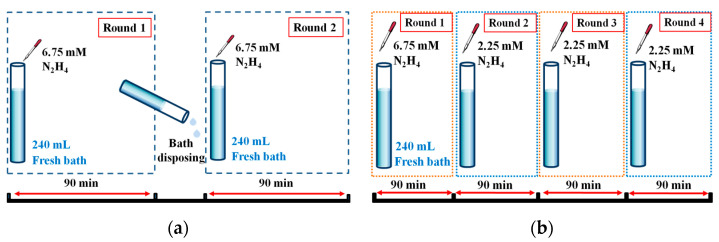
Schematics of the two approaches of hydrazine addition employed in this work: (**a**) one-time addition; (**b**) sequential addition.

**Figure 2 materials-14-04894-f002:**
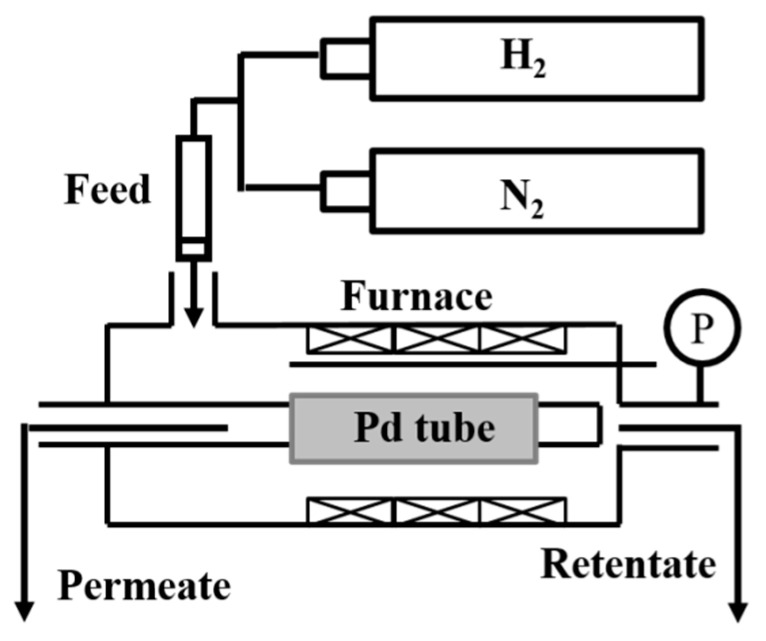
The homemade permeation-measuring module for pure hydrogen and nitrogen permeation measurements.

**Figure 3 materials-14-04894-f003:**
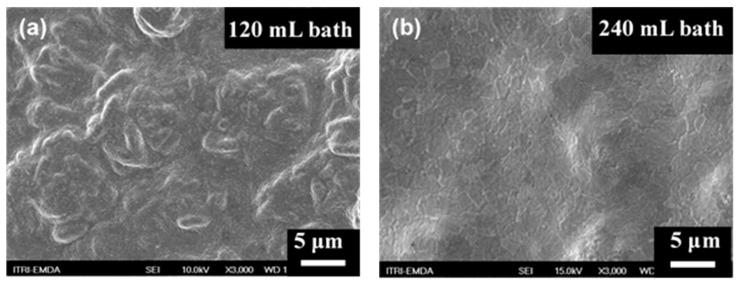
SEM plane view images of the membranes prepared using the same 6.75 mM concentration of the hydrazine addition but different volumes of bath: (**a**) 120 mL; (**b**) 240 mL.

**Figure 4 materials-14-04894-f004:**
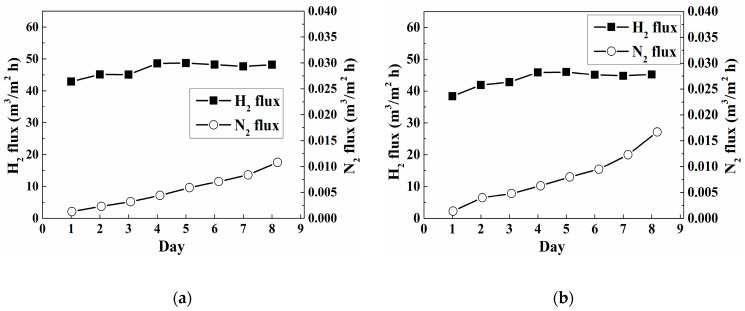
Thermal stability tests of the membranes were prepared using the same bath volume of 180 mL but different concentrations of hydrazine: (**a**) 6.75 mM; (**b**) 9 mM.

**Figure 5 materials-14-04894-f005:**
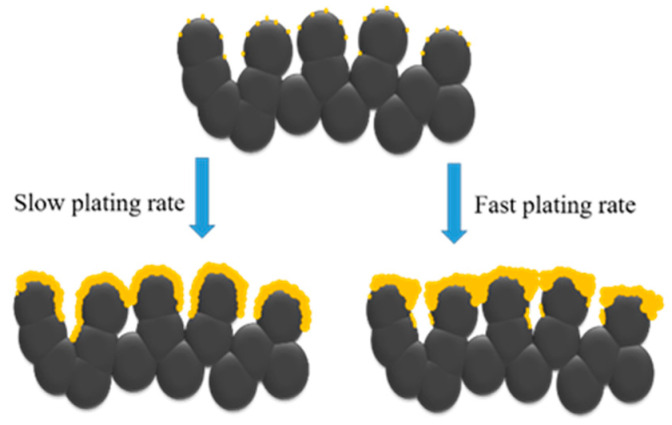
Schematics illustrating the stacking behaviors of Pd with slow and fast plating rates.

**Figure 6 materials-14-04894-f006:**
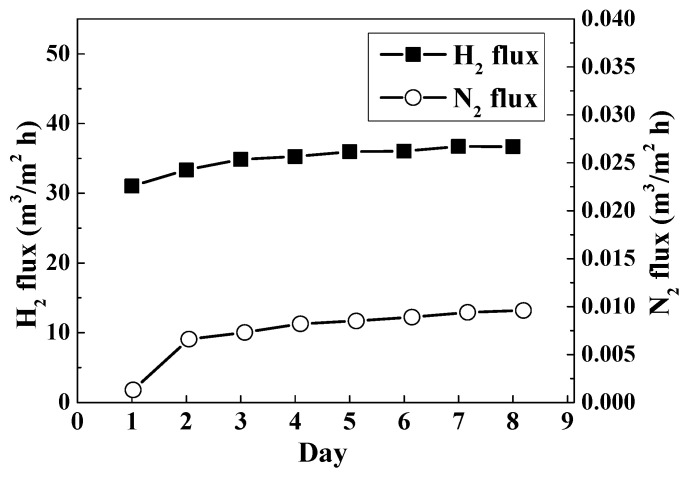
Thermal stability test of the Pd membrane prepared using the EDTA-free bath and the sequential addition of hydrazine.

**Table 1 materials-14-04894-t001:** Compositions of EDTA-free baths for electroless Pd plating.

Components	Amount
PdCl_2_	3.2 g/L
HCl	4 mL/L
NH_4_OH	320 mL/L

**Table 2 materials-14-04894-t002:** Properties and performances of various membranes prepared using different bath volumes and hydrazine concentrations.

N_2_H_4_ Conc. (mM)	Bath Volume (mL)	Plating Yield ^a^ (%)	Pd Thickness (μm)	H_2_ Permeance ^b^ (mol/m^2^ s Pa^0.5^)	N_2_ Flux ^c^ (mol/m^2^ s)	Selectivity ^d^ [H_2_/N_2_]
4.5	120	33	2.2	# ^e^	9.6 × 10^−1^	# ^e^
4.5	180	31	3.0	# ^e^	2.0 × 10^−3^	# ^e^
4.5	240	33	4.3	5.9 × 10^−3^	8.5 × 10^−5^	1.1 × 10^2^
6.75	120	55	3.6	6.8 × 10^−3^	1.9 × 10^−3^	1.7 × 10^2^
6.75	180	51	5.0	4.4 × 10^−3^	6.0 × 10^−6^	1.6 × 10^4^
6.75	240	46	6.0	3.4 × 10^−3^	1.3 × 10^−5^	5.9 × 10^3^
9	120	86	5.5	3.4 × 10^−3^	5.8 × 10^−5^	8.6 × 10^2^
9	180	60	5.8	3.1 × 10^−3^	9.6 × 10^−6^	5.0 × 10^3^
9	240	47	6.1	3.3 × 10^−3^	7.2 × 10^−6^	8.1 × 10^3^
13.5	120	93	6.0	3.3 × 10^−3^	2.2 × 10^−5^	1.3 × 10^3^
13.5	180	66	6.4	2.9 × 10^−3^	2.4 × 10^−6^	1.8 × 10^4^
13.5	240	56	7.2	2.6 × 10^−3^	4.8 × 10^−6^	1.1 × 10^4^
18	120	96	6.2	2.8 × 10^−3^	1.7 × 10^−3^	83
18	180	84	8.2	2.2 × 10^−3^	1.2 × 10^−4^	3.0 × 10^2^
18	240	68	8.9	2.0 × 10^−3^	3.8 × 10^−5^	1.5 × 10^3^

^a^ Ratio of estimated Pd^2+^ usage (derived from the weight gain after deposition) and the original Pd^2+^ in the bath. ^b^ Measured at 400 °C. ^c^ Measured at room temperature and 1-bar pressure difference prior to H_2_ flux measurements. ^d^ Defined as the ratio of the H_2_ flux to the N_2_ flux at 400 °C and 4-bar pressure difference. ^e^ Did not undergo the permeance measurements due to the large N_2_ leakage fluxes.

**Table 3 materials-14-04894-t003:** Compositions of EDTA-free baths for electroless Pd plating.

N_2_H_4_ Addition	Thickness	Yield	H_2_ Permeance	N_2_ Flux ^c^	Selectivity ^c^
	μm	%	mol/m^2^ s Pa^0.5^	mol/m^2^ s	H_2_/N_2_
One-Time ^a^	6.00	46	5.9 × 10^−3^	1.3 × 10^−5^	5.9 × 10^3^
Sequential ^b^	6.00	93	3.0 × 10^−3^	1.2 × 10^−6^	6.8 × 10^4^

^a^ Based on [N_2_H_4_] = 6.75 mM and plating bath volume = 240 mL, as schematically shown in Figure 1a. ^b^ Based on [N_2_H_4_] = 6.75, 2.25, 2.25, and 2.25 mM and plating bath volume = 240 mL, as schematically shown in Figure 1b. ^c^ The measurement conditions are the same as those in Table 1.

## Data Availability

Data are available on demand by asking the corresponding author.

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
