# Peer review of "Electroless Plating of High-Performance Composite Pd Membranes with EDTA-Free Bath"

_materials, 2021, doi:10.3390/ma14174894_

Round 1

Reviewer 1 Report

The manuscript deserves publication being a topical paper well organized and clearly paper with conclusion based on experimental results.

Before publication a revision needs to be operated to rewrite a better introduction explaining better the novel character with more references.. The manuscript has in the present version  only 20 references and all of them older than five years

Reviewer 2 Report

Dear Authors,

I appreciate the experimental effort you made for the study of “Electroless plating of high-performance composite Pd membranes with EDTA-free bath” using hydrazine as reducing egent.

In my opinion, some issues must be addressed before publication.

  • Lines 97 – 99 – please reformulate the phrase.
  • Please detail the phrase “After surface modification, Pd membranes were deposited onto the PSS tubes at 30 100 ºC with an EDTA-free bath as described in Table 1.” From the experimental section.
  • The plating bath solutions preparation is not clearly presented.
  • SEM experimental setup is not presented in the Materials and Methods section. How were the samples for SEM analysis prepared?
  • The affirmation “indicating a high possibility of fully 168 coverage of Pd on the PSS tube surface was obtained.” should be reconsidered. There are no experimental evidence to indicate this possibility.
  • The paragraph “The distinct morphologies of the two Pd membranes suggest that a threshold thickness is needed to achieve a dense Pd membrane. Mardilovich et al. have shown that there required a least Pd film thickness to form a dense membrane on a porous substrate by electroless plating, which is about three  times the size of the largest pore of the porous substrate.” should be reconsidered. The experimental data shows that according to Mandrilovich there is a threshold thickness achieved when the hydrazine volume ranged between 120 to 240 mL.
  • How was calculated the plating yield?
  • How was the Pd thickness appreciated? Please give experimental details for Pd thickness measurements.

Reviewer 3 Report

Manuscript materials-1333920 descibes interesting results of research on electroless plating of Pd membranes. It is a valuable research wjich might be interesting to many readers.

Despite this fact there are some remarks which should be concidered before publication of the manuscript in Materials.

1) line 174 Table 2, line 14 etc.
Values of obtained selectivities, e.g. 16128, seem to be very accurate. What is the measurement error? I would recommend to use scientific notation.

2) e.g. lines 13-14
Subscripts and superscripts are not used in many parts of the manuscirpt. It should be corrected as it is difficult to find real units.

3) Materials and Methods section
Manufacturer coutry and city should provided for each equipement and chemicals used in the investigation

4) line 237 Table 3
What is definition of yield?

5) line 16 - mistype - "6 ?m thick"

Round 2

Reviewer 2 Report

Dear Authors

I appreciate the efforts you made to make a clearer presentation of your work.

Reviewer 3 Report

Most of the remarks addressed during the first review were answered. Therefore, the manuscript may be recommended for publication.